# Evolutionary diversification of the trypanosome haptoglobin-haemoglobin receptor from an ancestral haemoglobin receptor

Harriet Lane-Serff[1†], Paula MacGregor[2†], Lori Peacock[3,4], Olivia JS Macleod[2], Christopher Kay[4], Wendy Gibson[4], Matthew K Higgins[1*], Mark Carrington[2*]

[1]Department of Biochemistry, University of Oxford, Oxford, United Kingdom; [2]Department of Biochemistry, University of Cambridge, Cambridge, United Kingdom; [3]School of Veterinary Science, University of Bristol, Bristol, United Kingdom; [4]School of Biological Sciences, University of Bristol, Bristol, United Kingdom

**Abstract** The haptoglobin-haemoglobin receptor of the African trypanosome species, *Trypanosoma brucei*, is expressed when the parasite is in the bloodstream of the mammalian host, allowing it to acquire haem through the uptake of haptoglobin-haemoglobin complexes. Here we show that in *Trypanosoma congolense* this receptor is instead expressed in the epimastigote developmental stage that occurs in the tsetse fly, where it acts as a haemoglobin receptor. We also present the structure of the *T. congolense* receptor in complex with haemoglobin. This allows us to propose an evolutionary history for this receptor, charting the structural and cellular changes that took place as it adapted from a role in the insect to a new role in the mammalian host.

*For correspondence: matthew.
higgins@bioch.ox.ac.uk (MKH);
mc115@cam.ac.uk (MC)

[†]These authors contributed
equally to this work

**Competing interests:** The
authors declare that no
competing interests exist.

**Reviewing editor:** Dominique
Soldati-Favre, University of
Geneva, Switzerland

## Introduction

Infection of livestock by African trypanosomes has a significant effect on food production in sub-Saharan Africa (*Shaw et al., 2004*). In contrast to human disease, which is caused by a restricted set of subspecies of *Trypanosoma brucei* (*Laveran, 1902*; *Pays and Vanhollebeke, 2009*), livestock disease is caused by at least six distinct species of African trypanosome, the most prevalent being *T. congolense* and *T. vivax* (*Rotureau and Van Den Abbeele, 2013*). While all share some common features, including antigenic variation and transmission by the tsetse fly, one of the most obvious differences between species is variation in the developmental cycle in the fly (*Hoare, 1972*) and in particular the location of the epimastigote developmental stage. *T. brucei* epimastigotes attach to the epithelium in the salivary glands away from the digestive tract, whereas *T. congolense* and *T. vivax* attach in the proboscis within the digestive tract (*Hoare, 1972*; *Peacock et al., 2012*; *Jefferies et al., 1987*). The basis for these different tissue tropisms is not known.

The cell surface of an African trypanosome acts as the molecular interface with its host, and the developmental transitions of the life cycle involve radical changes in cell surface composition, presumably as adaptations to different host niches. The best understood stage is the mammalian bloodstream form where the cell surface is covered with a dense layer of variant surface glycoprotein (VSG), which acts to protect the plasma membrane, enhancing survival of individual cells and allowing antigenic variation to ensure population survival (*Schwede and Carrington, 2010*; *Horn, 2014*). The density of packing of the VSG molecules on the surface of the bloodstream form trypanosome is thought to approach the maximum possible (*Grünfelder et al., 2002*). In contrast, the

**eLife digest** Trypanosomes are single-celled parasites that infect a range of animal hosts. These parasites need a molecule called haem to grow properly and are mostly spread by insects that feed on the blood of mammals. Most haem in mammals is found in red blood cells and is bound to a protein called haemoglobin. When it is released from these cells, haemoglobin forms a complex with another protein called haptoglobin as well.

The best-studied trypanosomes from Africa have a receptor protein on their surface that recognizes the haptoglobin-haemoglobin complex and allows the parasites to obtain haem from their hosts. An African trypanosome called *T. brucei* causes sleeping sickness in humans, and has a receptor that can only recognize haemoglobin when it is in complex with haptoglobin. However, few trypanosome receptors have been studied to date, and so it was not clear if they all work in the same way.

*Trypanosoma congolense* is a trypanosome that has a big impact on livestock farmers in sub-Saharan Africa and infects cattle, pigs and goats. Lane-Serff, MacGregor et al. now report that the receptor protein from *T. congolense* can bind to haemoglobin on its own. A technique called X-ray crystallography was used to reveal the three-dimensional structure of the *T. congolense* receptor and haemoglobin in fine detail. Further experiments then confirmed that the receptor actually binds more strongly to haemoglobin than it does to the haptoglobin-haemoglobin complex.

Experiments with living parasites showed that *T. congolense* produces its receptor when it is in the mouthparts of its insect host, the tsetse fly. This is unlike what occurs in *T. brucei*, which only produces its receptor while it is in the bloodstream of its mammalian host. Lane-Serff, MacGregor et al. suggest that *T. congolense*'s receptor is more like the receptor found in ancestor of the trypanosomes. This means that, at least once during the evolution of these parasites, this receptor evolved from being a haemoglobin receptor produced in the tsetse fly to a haptoglobin-haemoglobin receptor produced in an infected mammal.

The next step is to investigate the details of the role played by the *T. congolense* receptor when the parasite is in the tsetse fly. It will also be important to understand how this parasite is still able to grow in the mammalian host's bloodstream even though it does not produce much of the receptor during this stage.

developmental stages found inside insects, including the procyclic and epimastigote forms, have less densely packed surface coats, and contain different sets of surface proteins, including GARP in *T. congolense* and procyclins and BARP in *T. brucei* (*Bayne et al., 1993*; *Roditi et al., 1989*; *Urwyler et al., 2007*; *Beecroft et al., 1993*). Other cell surface proteins, including receptors and transporters, must operate in the context of these different cell surface architectures (*Lane-Serff et al., 2014*; *Stødkilde et al., 2014*).

The haptoglobin-haemoglobin receptor of *Trypanosoma brucei* (TbHpHbR) is the best character-ised trypanosome receptor. It is expressed in the mammalian bloodstream form and is used for the uptake of haptoglobin-haemoglobin complexes (HpHb) for haem acquisition (*Vanhollebeke et al., 2008*). In humans and some other primates, TbHpHbR also plays a role in innate immunity. Human serum contains two complexes, trypanolytic factors-1 and -2 (TLF1 and TLF2), which cause trypano-some lysis (*Rifkin, 1978*; *Hajduk et al., 1989*; *Tomlinson et al., 1995*; *Raper et al., 1996*). TLF1 and TLF2 both contain the apolipoprotein L1 toxin (*Vanhamme et al., 2003*) and a complex of haemo-globin bound to haptoglobin-related protein (HprHb) (*Raper et al., 1996*). It is the binding of HprHb to TbHpHbR that provides the uptake route for TLF1 into the trypanosome (*Vanhollebeke et al., 2008*).

High-resolution structures of TbHpHbR, both alone and in complex with HpHb, have shown how it can function within the densely packed VSG layer (*Lane-Serff et al., 2014*; *Stødkilde et al., 2014*). The N-terminal domain of TbHpHbR is formed from an extended three α-helical bundle with a small, membrane-distal head, and is attached to the plasma membrane by a glycophosphatidylinosotol-anchor at the C-terminus of a small C-terminal domain. HpHb binds along the membrane-distal half of the helical bundle. A striking feature of this helical bundle is a ~50° kink, which lies between the

HpHb binding site and the membrane attachment point. This kink is likely to result in separation of the VSG molecules on either side of the receptor, holding them apart and presenting the HpHb binding site to the extracellular environment. It also allows two receptors to contact a single HpHb dimer, allowing greater avidity for the ligand and more efficient uptake into trypanosomes (*Lane-Serff et al., 2014*). Specificity for HpHb results from direct, simultaneous contact between the receptor and both haptoglobin and the β-subunit of haemoglobin. Neither isolated haemoglobin nor haptoglobin binds significantly (*Vanhollebeke et al., 2008*). In addition, the propionate chains of haem directly contact the receptor and contribute to binding, with a significant reduction in binding affinity for HpHb that lacks haem (*Stødkilde et al., 2014*). Therefore TbHpHbR has evolved specific adaptations to function in the context of the VSG layer and to selectively bind to haem-loaded HpHb complexes.

The *T. congolense* receptor (TcHpHbR) was identified by sequence homology to the receptor from *T. brucei* and also binds to HpHb with low micromolar affinity (a $K_D$ of 8 μM for TcHpHbR compared with 1 μM for TbHpHb) (*Higgins et al., 2013*). Mutagenesis studies showed that TcHpHbR and TbHpHbR use overlapping binding sites to interact with HpHb (*Higgins et al., 2013*). The structure of TcHpHbR has a similar architecture to TbHpHbR, with a long three α-helical bundle and small membrane-distal head. However, one striking difference is that the helical bundle of TcHpHbR lacks the kink found in TbHpHbR. With the kink suggested to play a critical role in the operation of TbHpHbR in the context of the VSG layer, it was surprising that TcHpHbR lacks this evolutionary adaption if it too operates in a similar, densely packed environment. For this reason, we set out to investigate the structure and function of the *T. congolense* HpHbR, to understand its ligand binding specificity, its site of action and the evolutionary adaptations undergone by haptoglobin-haemoglobin receptors from different species.

## Results

### The *T. brucei* haptoglobin-haemoglobin receptor has evolved from a haemoglobin receptor

To gain an understanding of ligand binding by TcHpHbR, we aimed to determine its structure in complex with HpHb. Human haemoglobin in complex with the SP domain of human haptoglobin (HpSP) was mixed with a small molar excess of TcHpHbR prior to gel filtration chromatography, in order to purify receptor-ligand complex. However, analysis of the resulting fractions from the chromatography column revealed a higher molecular weight peak containing TcHpHbR and Hb, but no HpSP. A lower molecular weight peak, at the elution volume expected for unliganded TcHpHbR, contained excess free TcHpHbR and HpSP (*Figure 1A*). Therefore the presence of TcHpHbR led to disassembly of the HpSPHb complex and the formation of a complex containing TcHpHbR and Hb.

While native haptoglobin is cleaved to form α- and β-subunits, the recombinant HpSP used in this study is not cleaved. We therefore repeated the gel filtration experiment using native Hp in an HpHb complex. In this case, we found that the HpHb complex is not disassembled by incubation with the receptor, suggesting that the receptor does not disassemble native HpHb complexes during its physiological function (*Figure 1—figure supplement 1A*). However, the complex that it forms with the receptor is not sufficiently strong to remain intact during gel filtration chromatography and HpHb and the receptor elute in separate peaks, confirming that the receptor has a low affinity for HpHb.

The discovery that TcHpHbR binds to Hb was unexpected as TbHpHbR had previously been shown to bind HpHb but not to free Hb (*Vanhollebeke et al., 2008*). Surface plasmon resonance (SPR) was therefore used to investigate the ligand specificity of HpHbRs from *T. brucei, T. congolense* and *T. vivax*. The receptors were immobilised on the chip surface through a C-terminal biotin moiety, causing them to be presented in the same orientation as on the cell surface. Binding was then measured for both Hb and HpHb. As expected, HpHbR from *T. brucei* interacted with HpHb, but not with Hb alone (*Figure 1B*). However, HpHbRs from both *T. congolense* and *T. vivax* interacted with either HpHb or free Hb. Indeed, both receptors formed a far more stable complex with Hb than with HpHb, as shown by the lower off-rate (*Figure 1B*). In addition when Hb was mixed with TcHpHbR and subjected to gel filtration chromatography, the primary peak was a complex of

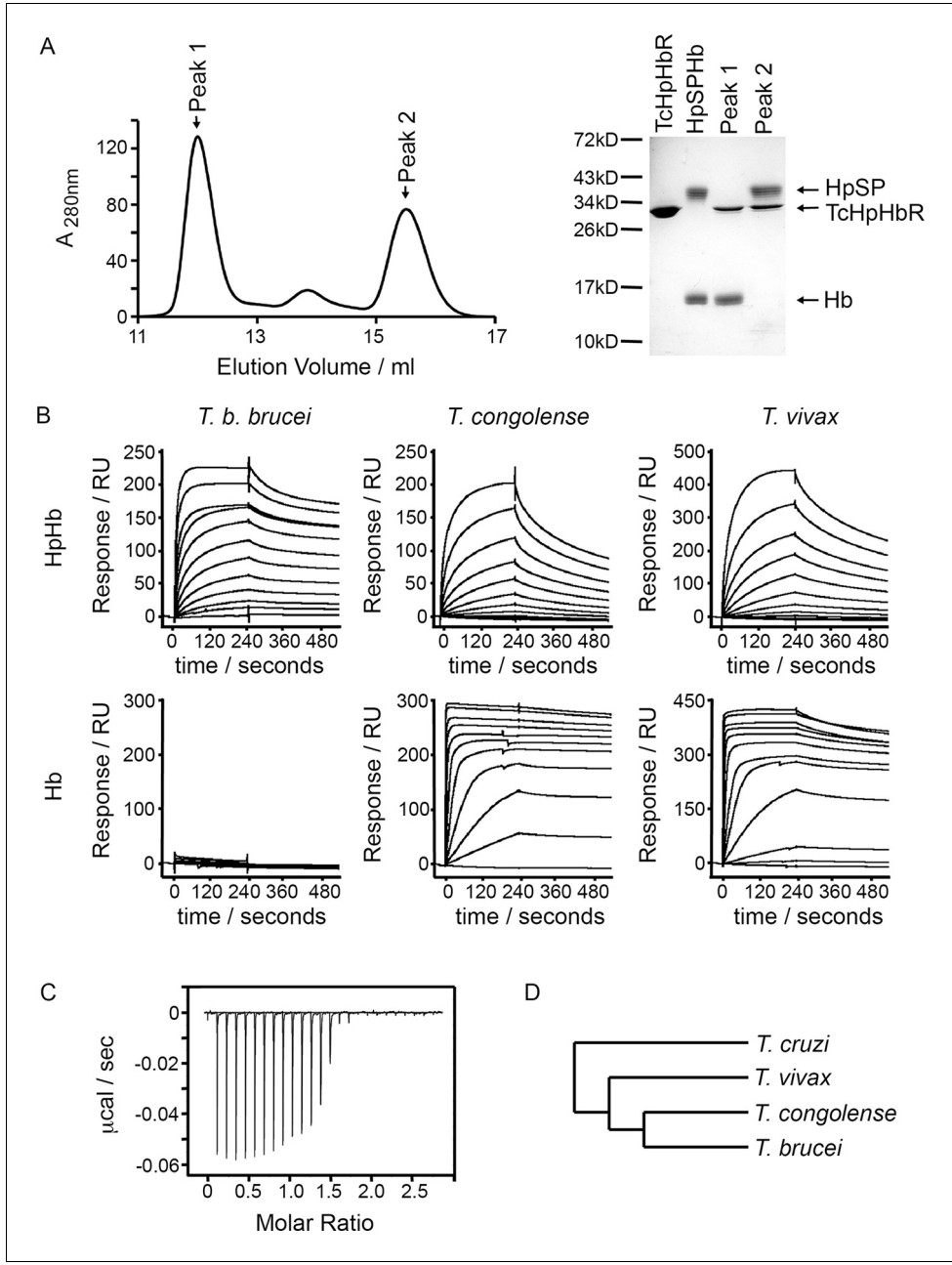

**Figure 1.** *T. congolense* and *T. vivax* HpHbRs are haemoglobin receptors. (**A**) Analytical gel filtration chromatography analysis showing the consequence of mixing HpSPHb and TcHpHbR. A mixture of TcHpHbR and HpSPHb was loaded onto the column, resulting in two peaks. The gel shows that peak 1 contains a complex of TcHpHbR bound to Hb, while peak 2 contains free TcHpHbR and free HpSP. (**B**) Surface plasmon resonance analysis of the binding of HpHbRs from *T. b. brucei*, *T. congolense* and *T. vivax* to either Hb or HpHb. (**C**) Isothermal titration calorimetry shows the binding of two TcHpHbR to one Hb tetramer. (**D**) A phylogenetic tree indicating the evolutionary history of the trypanosome strains under study (***Stevens et al., 2001***).

The following figure supplements are available for figure 1:

**Figure supplement 1.** TcHpHbR does not disassemble HpHb.

**Figure supplement 2.** TcHpHbR binding to bovine Hb.

TcHpHbR bound to Hb, indicating that TcHpHbR forms a more stable complex with Hb than with HpHb (*Figure 1—figure supplement 1B*).

*T. congolense* is not a human infective pathogen, but is found in numerous livestock species. We therefore also tested the binding of TcHpHbR to bovine haemoglobin and observed a strong interaction with a slow off rate (*Figure 1—figure supplement 2*). Therefore TcHpHbR has a high affinity for Hb, while TbHpHbR does not bind to Hb alone. A similar change in specificity is seen in the mammalian scavenger receptor CD163, as mouse CD163 binds to Hb while human CD163 binds to HpHb alone (*Etzerodt et al., 2013*).

As haemoglobin is a symmetrical tetramer of two α and two β subunits, each tetramer could potentially bind to two receptors and the SPR measurements described above would result from a mixture of monovalent and bivalent binding. Isothermal titration calorimetry (ITC) was therefore used to measure the monovalent $K_D$ and the stoichiometry of the interaction between TcHpHbR and Hb (*Figure 1C*). This revealed that two receptors interact with one Hb tetramer. This follows the same pattern as the TbHpHbR:HpHb complex, where two receptors bind to each dimeric HpHb complex (*Lane-Serff et al., 2014*; *Stødkilde et al., 2014*). The $K_D$ for the TcHpHbR:Hb interaction was estimated to be 3 nM by ITC. This is around 1000-fold tighter than the affinity of the same receptor for HpHb, previously measured by ITC to be 3 µM (*Higgins et al., 2013*).

*T. congolense* and *T. brucei* diverged from each other after their last common ancestor had diverged from *T. vivax* (*Figure 1D*) (*Stevens et al., 1999*; *Hamilton et al., 2004*; *Kelly et al., 2014*). As HpHbRs from both *T. congolense* and *T. vivax* bind to Hb, it is most likely that this property was lost from *T. brucei* after it diverged from *T. congolense* rather than separately gained in both *T. congolense* and *T. vivax*. The ~1000-fold higher affinity of TcHpHbR for Hb than for HpHb also suggests that Hb binding is the major evolved function of this receptor in *T. congolense* and *T. vivax*.

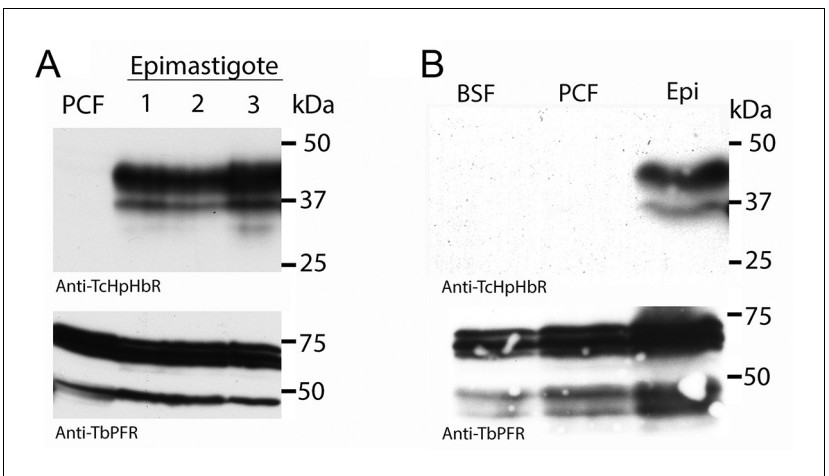

**Figure 2.** Western analysis reveals high levels of HpHbR expression in epimastigote-enriched cultures. (**A**) Three populations of *T. congolense* epimastigotes were generated in vitro by maintaining procyclic form cultures of *T. congolense* IL3000 cells at stationary phase (*Coustou et al., 2010*). The majority of epimastigote forms are adherent and stick to the culture flask while the culture supernatant is predominantly trypomastigotes with a low percentage of detached epimastigote forms. Cell lysates where generated from culture supernatants and subject to western blot analysis. No TcHpHbR protein expression was detected in procyclic form cultures (PCF) whereas TcHpHbR expression was observed in all epimastigote-containing cultures. The protein is observed above the expected 32 kDa, probably due to the GPI-anchor and N-glycosylation affecting mobility as has been observed for the TbHpHbR (*Vanhollebeke et al., 2008*). Loading control is anti-TbPRF2. (**B**) No TcHpHbR protein expression was detected in *T. congolense* bloodstream forms (BSF) or procyclic form cultures (PCF) by western blot, whereas expression was detected in epimastigote-containing cultures (Epi).

The following figure supplement is available for figure 2:

**Figure supplement 1.** Validation of TcHpHbR antisera and quantification of TcHpHbR protein expression in *T. congolense* epimastigotes.

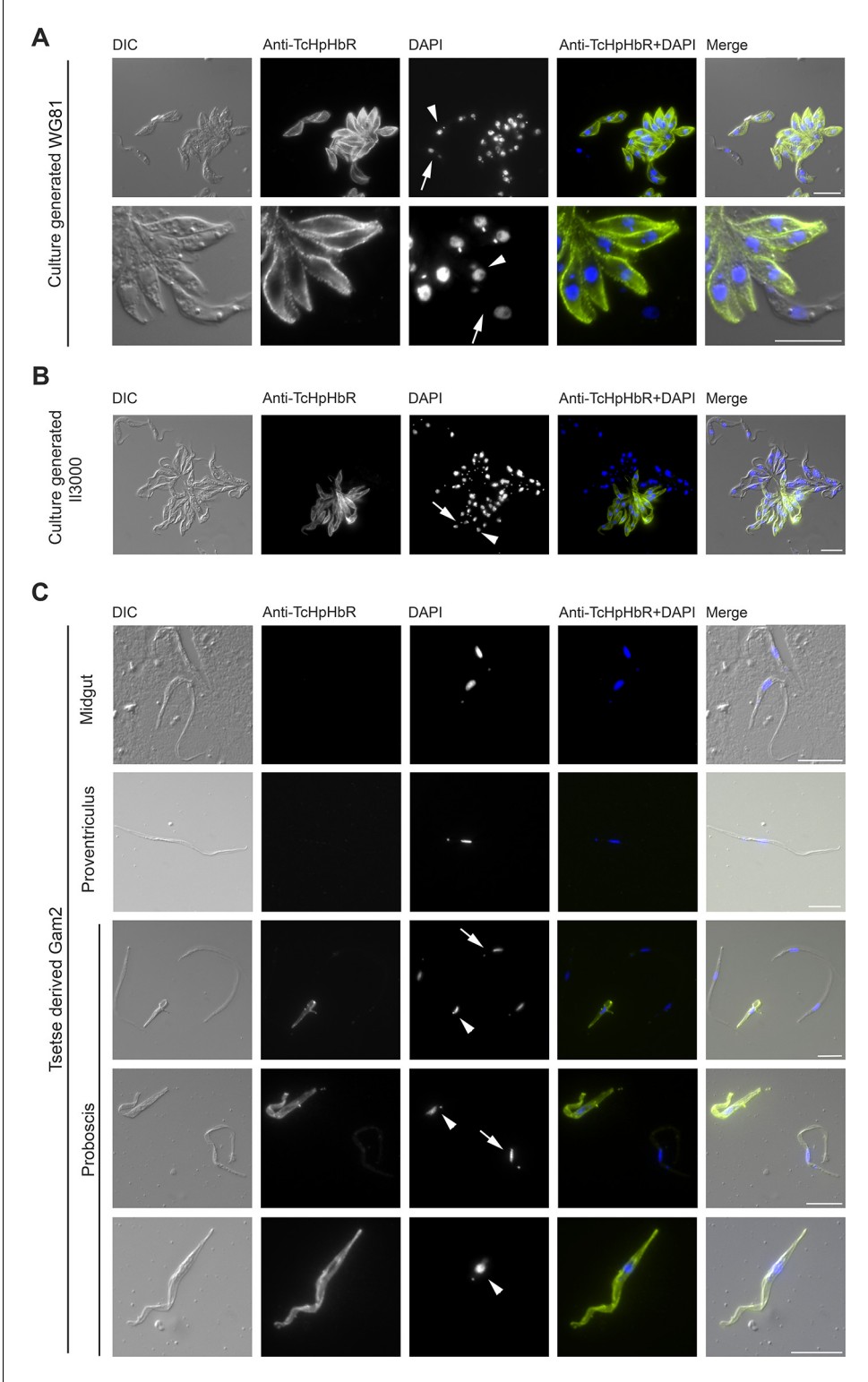

**Figure 3.** Immunofluorescent analysis of *T. congolense* epimastigotes reveals TcHpHbR protein is expressed at high levels across the entire cell surface. (**A**) Immunofluorescence analysis of paraformaldehyde-fixed (non-permeabilised) in vitro generated *T.congolense* WG81 epimastigotes with rabbit anti-TcHpHbR antisera and an Alexa488 conjugated anti-rabbit secondary antibody. TcHpHbR was readily detected on the surfaces of all cells where kinetoplast repositioning had occurred. (Arrow highlights a trypomastigote, arrowhead highlights an epimastigote) (**B**) This was also observed with immunofluorescence analysis of methanol-fixed (permeabilised) in

*Figure 3 continued on next page*

*Figure 3 continued*

vitro generated *T. congolense* Il3000 epimastigotes. Occasionally TcHpHbR expression was detected in cells that did not display kinetoplast repositioning. (Arrow highlights a TcHpHbR positive cell without associated kinetoplast repositioning.) (C) *T.congolense* Gam2 cells were harvested from the midgut (top panel), proventriculus (second panel) and proboscis (lower three panels) of tsetse flies. Cells were fixed with methanol and immunofluorescence analysis was carried out as above. Trypanosomes harvested from the midgut (top panel) were always negative for TcHpHbR and those harvested from the proventriculus (second panel) were mostly negative for TcHpHbR, although occasional cells were identified with a faint positive signal. Epimastigotes harvested from the proboscis (lower three panels, arrowheads) were always strongly positive for TcHpHbR. Trypomastigotes from the proboscis (lower three panels, arrows) showed a faint or negative signal for TcHpHbR (arrows). All scale bars represents 10 μm.

The following figure supplement is available for figure 3:

**Figure supplement 1.** TcHpHbR is expressed on a cell undergoing asymmetric division.

Together these observations suggest that the ancestor of the HpHbRs was primarily a haemoglobin receptor and that evolutionary changes that have taken place during the evolution of *T. brucei* have led to an alteration in its binding specificity.

## The haptoglobin-haemoglobin receptor of *T. congolense* is expressed in the epimastigote developmental stage

The finding that TcHpHbR is a haemoglobin receptor was unexpected in the light of our current knowledge of TbHpHbR. In *T. brucei*, the receptor is expressed in the mammalian bloodstream stage of the parasite, where it functions in the acquisition of haem as a nutrient (*Vanhollebeke et al., 2008*). In mammalian blood, free haemoglobin levels are extremely low due to the presence of haptoglobin as a scavenger. Haptoglobin binds free Hb, allowing it to be removed

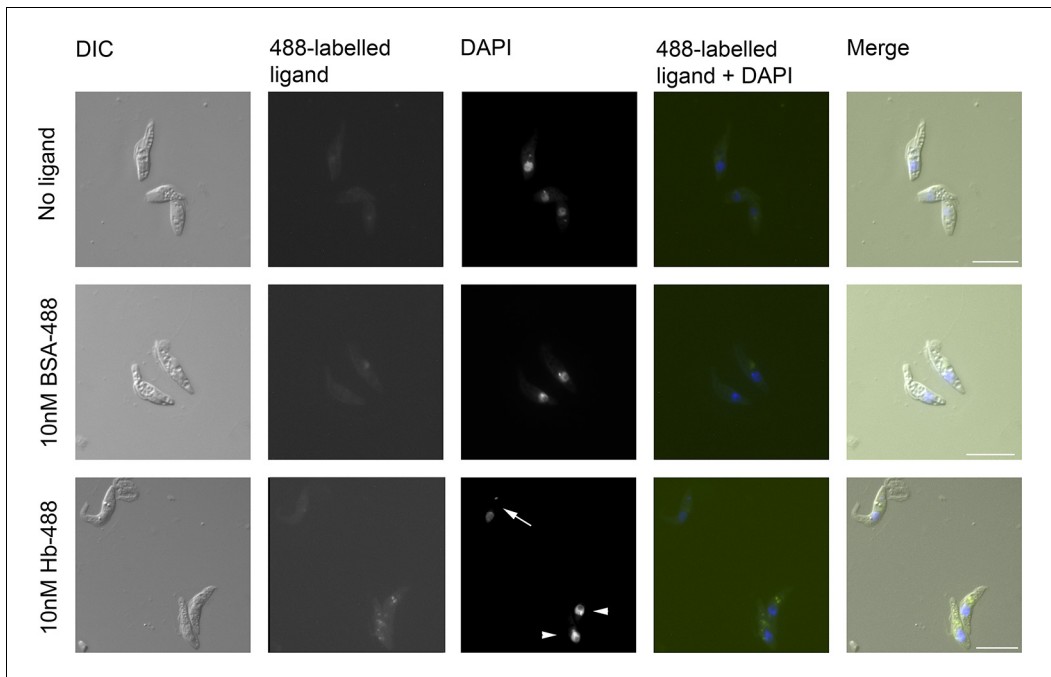

**Figure 4.** *T. congolense* epimastigotes internalise 488-labelled Hb. Assay for uptake of Alexa488-labelled Hb (Hb-488) and Alexa488-labelled BSA (BSA-488) into *T. congolense* WG81 epimastigotes was monitored by microscopy. Uptake of Hb-488 was detected at 10 nM in epimastigotes (lower panel, arrowheads) but not in trypomastigotes (lower panel, arrow). No fluid phase uptake of BSA-488 at 10 nM in any cells (centre panel).

from the serum by endocytosis of HpHb complexes into macrophages, reducing the potential for oxidative damage caused by haem (*Kristiansen et al., 2001*). Therefore, except under conditions of exceptional haemolysis, there is little haemoglobin present in the blood, bringing into question the requirement for a haemoglobin receptor. We therefore assessed the life cycle stage of *T. congolense* in which the receptor is expressed.

While *T. congolense* will not be exposed to free haemoglobin in the mammalian bloodstream, developmental forms in the tsetse fly will be exposed to haemoglobin derived from the bloodmeal. A previous proteomic analysis comparing different developmental stages of *T. congolense* did not detect TcHpHbR protein in bloodstream forms but did detect it as an abundant protein in epimastigotes (*Eyford et al., 2011*). In addition, a transcriptome analysis of various developmental forms of *T. vivax* indicated that TvHpHbR mRNA was most abundant in epimastigotes (*Jackson et al., 2015*). To investigate further, *T. congolense* epimastigotes were generated from procyclic form cultures (*Coustou et al., 2010*). Western blot analysis of three independently generated epimastigote-containing cultures showed expression of the TcHpHbR in these populations, while expression was below detectable levels in the original procyclic cells or in bloodstream form *T. congolense* (*Figure 2*; *Figure 2—figure supplement 1*). Therefore differentiation to the epimastigote form was associated with expression of TcHpHbR.

Immunofluorescence analysis (IFA) was carried out next to determine the sub-cellular localisation of TcHpHbR using epimastigote forms generated in vitro from procyclic form cultures. The key

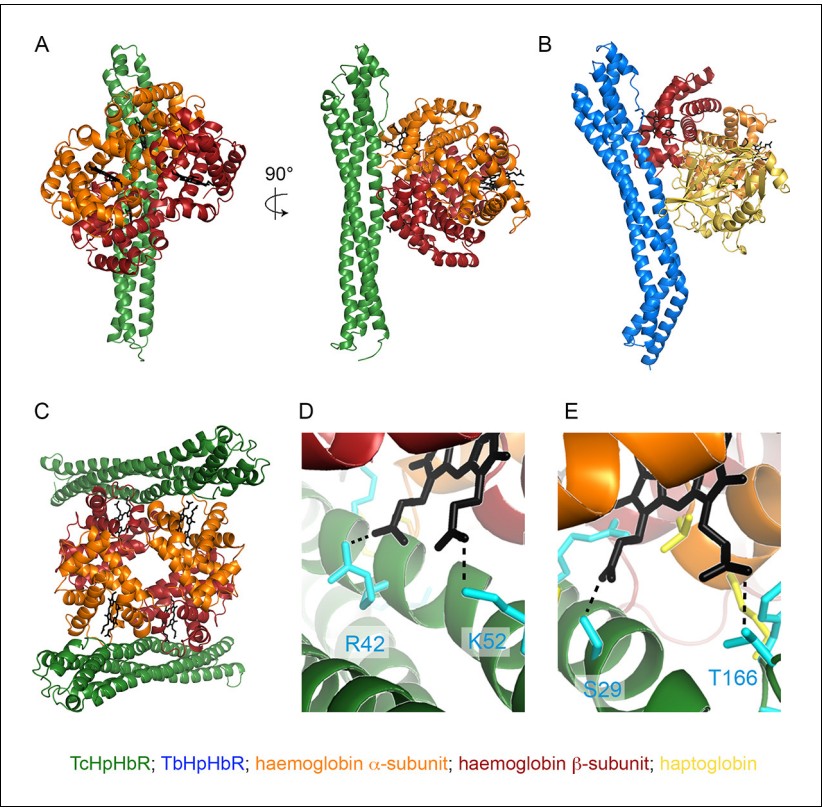

TcHpHbR; TbHpHbR; haemoglobin α-subunit; haemoglobin β-subunit; haptoglobin

**Figure 5.** The structural basis of haemoglobin binding by TcHpHbR. (**A**) The structure of a complex of TcHpHbR bound to Hb. (**B**) The structure of the TbHpHbR:HpSPHb complex (*Lane-Serff et al., 2014*). (**C**) The contents of the asymmetric unit of the TcHpHbR:Hb crystals, showing two receptors binding to a single haemoglobin tetramer. (**D**) A close up view of the interaction of TcHpHbR with the β-chain of haemoglobin showing the direct contacts made with the haem group. (**E**) A close up view of the interaction of TcHpHbR with the α-chain of haemoglobin showing the direct contacts made with the haem group.
The following figure supplement is available for figure 5:

**Figure supplement 1.** TcHpHbR-binding residues are conserved across mammalian haemoglobins.

distinguishing feature resulting from differentiation to epimastigotes is a change in the relative positions of the nucleus and kinetoplast along the anterior-posterior axis of the cell. In trypomastigote forms, such as procyclics, the kinetoplast is positioned posterior to the nucleus while in epimastigote forms it is juxtanuclear or anterior to the nucleus (*Peacock et al., 2012*; *Vickerman, 1984*). In all cells in which kinetoplast repositioning to that found in the epimastigote form had occurred, TcHpHbR protein was detected and localised across the whole cell surface, including the flagellum, in both permeabilised and non-permeabilised cells (*Figure 3A and B*).

A minority of cells that expressed TcHpHbR did not have the characteristic kinetoplast and nuclear positioning characteristic of epimastigotes (*Figure 3B*, arrow). This observation may indicate that: (i) TcHpHbR expression in epimastigotes occurs immediately prior to kinetoplast repositioning, and/or (ii) epimastigote formation in vitro is incomplete and/or (iii) the epimastigotes were differentiating further into metacyclic forms and that these, at least initially, retain TcHpHbR expression. The last possibility is supported by the identification of a TcHpHbR-expressing cell undergoing asymmetric division, where one daughter cell will be an epimastigote and the other will not, as occurs during metacyclogenesis in *T. brucei* (*Rotureau and Van Den Abbeele, 2013*) (*Figure 3—figure supplement 1*) and by the proteomic detection of low levels of TcHpHbR in metacyclic populations (*Eyford et al., 2011*).

To confirm that the TcHpHbR is indeed an epimastigote-stage protein in vivo, *Glossina pallidipes* tsetse flies were infected with *T. congolense* Gam2. Trypanosomes were harvested from the midgut, proventriculus and proboscis of infected flies 40 days post infection and expression of TcHpHbR was investigated by immunofluorescence. Trypomastigote forms harvested from the midgut (procyclics) were all negative for TcHpHbR staining (*Figure 3C*, top panel). Trypomastigotes from the proventriculus were also typically negative for TcHpHbR staining (*Figure 3C*, second panel), although some cells were identified with faint positive signal (data not shown). Trypanosomes collected from the tsetse proboscis included both trypomastigotes (proventricular trypomastigotes, pre-metacyclics or metacyclic forms) and epimastigotes (*Figure 3C*, lower three panels, arrows highlight trypomastigotes and arrowheads highlight epimastigotes). All epimastigotes identified had high levels of TcHpHbR expression, whereas trypomastigotes were either negative or

**Table 1.** Crystallographic statistics.

| Beamline | Diamond I03 |
|---|---|
| Space Group | $P22_12_1$ |
| Cell parameters (Å) | a=72.75, b=127.3, c=172.42 |
| Resolution (Å) | 101.89-3.2 |
| Wavelength (Å) | 0.976 |
| $R_{PIM}$ (%) | 7.3 (41.0) |
| I/ σ(I) | 7.3 (2.4) |
| Completeness (%) | 99.0 (99.1) |
| Multiplicity | 2.7 (3.0) |
| | |
| Resolution (Å) | 3.2 |
| No. reflections | 25191 |
| $R_{work}$ / $R_{free}$ (%) | 20.44 / 23.51 |
| No. of protein residues in model | 1063 |
| rmsd bond lengths (Å) | 0.010 |
| rmsd bond angles (°) | 1.21 |
| Ramachandran plot | |
| Preferred region | 93.1% |
| Allowed region | 6.9% |
| Outliers | 0% |

**Table 2.** Description of interactions between TcHpHbR and Hb.

| *TcHpHbR* | | Hb | | | | |
|---|---|---|---|---|---|---|
| Residue | Group | Chain | Residue | Group | | Interaction |
| | | Hbα | | | | |
| S29 | backbone CO | C/E | H46 | side chain | | Hydrogen bond |
| S29 | side chain | C/E | Haem | O1 | | Hydrogen bond |
| I30 | side chain | C/E | Patch | | | Hydrophobic |
| R37 | side chain NH1/NH2 | C/E | L92 | backbone CO | | Hydrogen bond |
| K127 | side chain | C/E | P45 | backbone CO | | Hydrogen bond |
| K130 | side chain | C/E | Patch | | | Hydrophobic |
| T166 | side chain | C/E | Haem | O3 | | Hydrogen bond |
| Y168 | side chain | C/E | Patch | | | Hydrophobic |
| Y168 | backbone CO | C/E | K91 | side chain | | Hydrogen bond |
| D169 | side chain | C/E | K91 | side chain | | Salt bridge |
| | | Hbβ | | | | |
| I41 | side chain | D/F | Patch | | | Hydrophobic |
| R42 | side chain NH2 | D/F | Haem | O1 | | Hydrogen bond |
| A44 | side chain | D/F | Patch | | | Hydrophobic |
| T45 | side chain | D/F | Patch | | | Hydrophobic |
| E47 | side chain OE2 | D/F | K96 | side chain | | Salt bridge |
| F48 | side chain | D/F | Patch | | | Hydrophobic |
| K52 | side chain | D/F | Haem | O3 | | Hydrogen bond |

weakly positive (*Figure 3C*, lower three panels). Therefore, TcHpHbR is highly expressed in the *T. congolense* epimastigote life-stage in vivo, with some upregulation of expression occurring prior to kinetoplast repositioning.

TcHpHbR expression was readily detected over the entire cell surface of epimastigotes, suggesting expression levels were higher than those of receptors previously characterized in bloodstream forms of *T. brucei*. The average copy number of TcHpHbR was therefore estimated using western blots to compare cell lysates from in vitro generated epimastigotes with known quantities of recombinant protein, with an adjustment for the percentage of cells in the culture expressing the protein as determined by immunofluorescence. This suggested an average of ~5–9 x $10^5$ TcHpHbR molecules to be present per TcHpHbR-expressing cell (*Figure 2—figure supplement 1*). For comparison, *T. brucei* bloodstream forms express approximately 200 to 400 TbHpHbR molecules per cell (*Vanhollebeke et al., 2008*; *Drain et al., 2001*). Therefore the *T. congolense* HpHbR is an abundant protein expressed in epimastigotes, with around a 1000-fold more receptors per cell than are found in the *T. brucei* bloodstream form.

## *T. congolense* epimastigotes internalise haemoglobin

To determine if TcHpHbR functioned in receptor-mediated endocytosis of Hb, ligand uptake was monitored in a live cell assay using culture-derived epimastigotes of *T. congolense* WG81. These cultures contained both trypomastigotes and epimastigotes. The trypanosomes were incubated with either 10 nM Alexa488-labelled Hb or 10 nM Alexa488-labelled BSA. Internalisation of Hb, but not BSA, was observed specifically in epimastigote forms but not in trypomastigote forms (*Figure 4*). Therefore, the *T. congolense* developmental form that highly expresses the TcHpHbR on its surface is indeed able to internalise Hb at low nanomolar concentrations.

## The structural basis for haemoglobin binding

To investigate the molecular basis for haemoglobin binding by TcHpHbR, we mixed the receptor with HpSPHb. As described above (*Figure 1A*), this generated a complex of TcHpHbR bound to Hb.

This complex crystallised with a well solution of 8% PEG 8000, 0.1 M sodium citrate pH 5.0 in 18 hr, and a complete data set was collected to 3.2 Å resolution. A molecular replacement solution was determined using *T. congolense* HpHbR (pdb: 4E40) and human Hb (pdb: 1HHO) as search models. This allowed placement of two receptor molecules and a single haemoglobin tetramer in the asymmetric unit of the crystal (*Figure 5*, *Table 1*). Each receptor makes the same interactions with haemoglobin, and the receptor conformation is unaltered from that of unliganded receptor (*Higgins et al., 2013*), with a root-mean-square deviation of 0.6 Å. This lack of conformational change on ligand binding matches that seen in the structures of *T. brucei* HpHbR alone and in the presence of HpHb (*Lane-Serff et al., 2014*). The haemoglobin is in the oxygenated conformation with a root-mean-square deviation of 0.7 Å from the search model.

The interaction surface can be divided into two subsites, with interactions made with the α-subunit of one haemoglobin dimer and the β-subunit of the second haemoglobin dimer (*Figure 5A*). This suggests that the receptor either binds selectively to haemoglobin tetramers or that two haemoglobin dimers will assemble together with the receptor. All receptor-ligand interactions are mediated by features that are absolutely conserved between haemoglobin from human and livestock

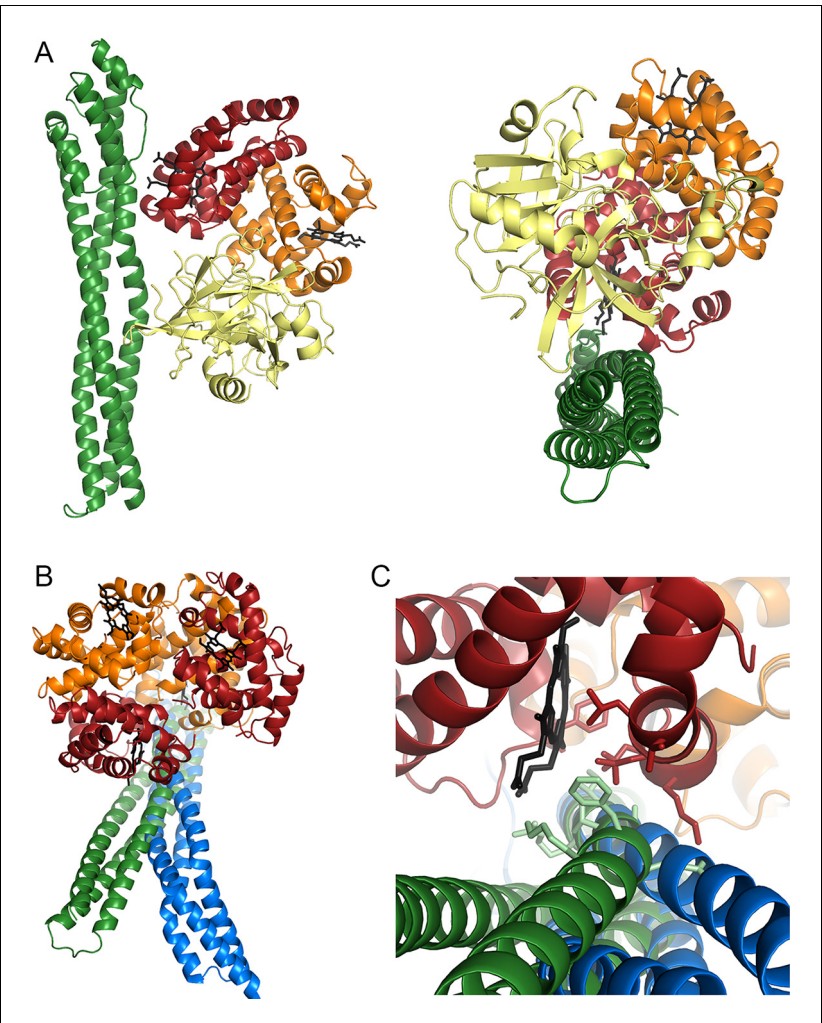

**Figure 6.** Understanding HpHbR ligand specificity. (**A**) A model of TcHpHbR bound to HpSPHb, based on the TcHpHbR:Hb structure. (**B, C**) The TcHpHbR:Hb and TbHpHbR:HpSPHb complexes have been aligned, with the haemoglobin subunit that interacts with the membrane distal binding site used for the alignment. This shows that a change in the path of the helical bundle of TbHpHbR (blue) prevents the interaction that occurs between TcHpHbR (green) and the membrane proximal haemoglobin subunit. This disruption of the membrane proximal binding site has caused TbHpHbR to lose affinity for Hb

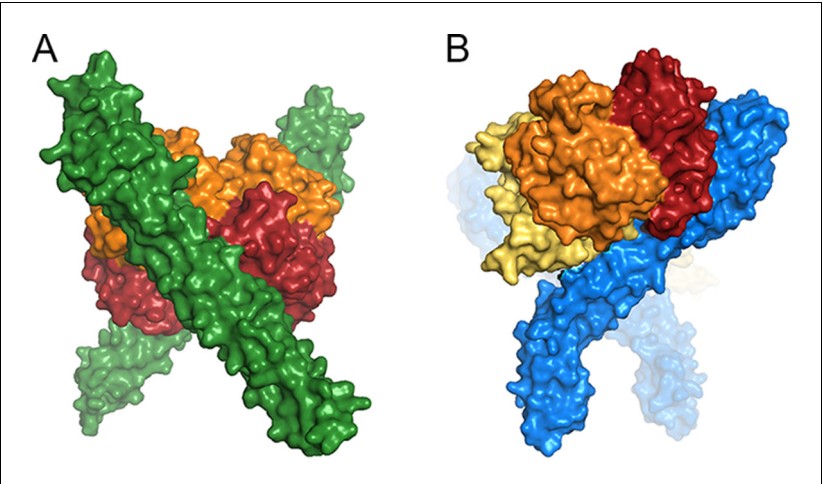

**Figure 7.** A comparison of ligand binding by HpHbRs from different species. Space filling models of (**A**) Two TcHpHbR molecules binding to a single haemoglobin tetramer. (**B**) Two TbHpHbRs bound to a single haptoglobin-haemoglobin dimer.

species (**Table 2**, **Figure 5—figure supplement 1**). It is therefore highly likely that the haemoglobin molecules of most of the mammals bitten by tsetse flies will bind to the receptor. The membrane-distal binding site interacts with the α-subunit of haemoglobin (**Figure 5E**). This contains more than half of the total interaction surface area ($\sim 735$ Å$^2$ of 1385 Å$^2$). It is predominantly mediated by hydrogen bonds and electrostatic contacts, with a significant component from a direct interaction between the receptor and the propionate chains of haem. This binding site is similar in location, size and chemical nature to the membrane-distal binding site of TbHpHbR for HpHb, in which the same region of TbHpHbR makes direct contacts with the β-subunit of haemoglobin (**Figure 5B**).

The smaller, membrane-proximal part of the binding site contacts the β-subunit of haemoglobin with a total contact surface area of 650 Å$^2$ (**Figure 5D**). This interface is more hydrophobic in nature than the membrane-distal site. However, once again, the propionate chains of haem directly interact with the receptor. This membrane distal binding site is distinct in position and chemical nature from the region of TbHpHbR that contacts haptoglobin, which is closer to the membrane surface and is smaller ($\sim 505$ Å$^2$). The more extensive contact site between TcHpHbR and the β-subunit of Hb, than that between TbHpHbR and Hp, is likely to be a major contributor to the increased affinity of TcHpHbR for Hb.

TcHpHbR also interacts with HpHb, albeit with a significantly lower affinity. To understand the molecular basis for this interaction, we generated a model of the TcHpHbR:HpHb complex (**Figure 6A**). One surprise from a comparison of the TcHpHbR:Hb structure with that of TbHpHbR: HpHb is that the membrane distal binding site of TcHpHbR interacts with the Hb α-subunit while the equivalent site of TbHpHbR interacts with the Hb β-subunit of HpHb. We modeled the TcHpHbR: HpHb structure by docking either Hb subunit from the structure of HpHb onto the interacting Hb α-subunit of the TcHpHbR:Hb complex. When the Hb α-subunit is docked onto the membrane-distal binding site, Hp makes no contacts with the receptor. In contrast, when the HpHb is 'flipped' so that the Hb β-subunit contacts the membrane-distal binding site, this positions Hp adjacent to the receptor (**Figure 6A**). In addition, the residues from the Hb α-subunit that contact the membrane-distal binding site are replaced by chemically similar residues in the Hb β-subunit, allowing equivalent interactions to occur. The interaction between TbHpHbR and Hp is predominantly hydrophobic in nature, and a similar, but smaller, hydrophobic region in TcHpHbR, involving phenylalanine 48 is similarly positioned to interact with Hp. We therefore predict that HpHb interacts with TcHpHbR with a similar binding mode to that observed in the TbHpHbR:HpHb complex with the Hb β-subunit contacting the membrane-distal binding site, allowing haptoglobin to bind to a membrane-proximal site.

Finally, a comparison of the TcHpHbR:Hb structure with that of TbHpHbR:HpHb allows us to rationalise the changes have taken place in TbHpHbR to cause the loss of Hb binding. While an alignment of these structures shows little alteration in the membrane-distal binding site, the path of helix I of the receptor is altered in TbHpHbR so that it no longer forms the membrane-proximal binding site for the Hb β-subunit, thereby reducing the total interaction surface area by nearly half (*Figure 6B and C*). This change in the helical pathway could be a consequence of the adoption of a kink in TbHpHbR, which allows the receptor to function in the context of the VSG layer. Alternatively, it could be a result of changes that lead to the increase in the affinity of the receptor for HpHb, from 8 μM in *T. congolense* to 1 μM in *T. brucei,* as determined by SPR (*Higgins et al., 2013*).

## Discussion

The haptoglobin-haemoglobin receptor of *T. brucei* has been extensively characterised because of its role in the uptake of haptoglobin-haemoglobin into the bloodstream form parasite (*Vanhollebeke et al., 2008*; *Lane-Serff et al., 2014*). This receptor also mediates uptake of trypanolytic factor-1 (*Vanhollebeke et al., 2008*) and to a lesser extent the uptake of trypanolytic factor-2 (*Bullard et al., 2012*), contributing to the inability of most isolates of *T. brucei* to survive in human serum. It was a reasonable expectation that orthologues in other African trypanosome species, such as *T. congolense* and *T. vivax*, would have a similar function. However, here it is shown that there are significant differences between the receptors from *T. brucei* and *T. congolense*. Firstly, *T. congolense* HpHbR has an approximately 1000-fold greater affinity for haemoglobin than for haptoglobin-haemoglobin and developmental forms expressing TcHpHbR are able to internalise Hb at low nanomolar concentrations. Secondly, the *T. congolense* receptor is expressed in the epimastigotes with a copy number approximately 1000-fold greater than that of *T. brucei* HpHbR in the bloodstream form. Finally, *T. congolense* HpHbR is distributed over the whole cell surface, whereas in *T. brucei* it is concentrated in the flagellar pocket (*Vanhollebeke et al., 2008*). Similar findings are seen for *T. vivax*, as TvHpHbR also binds haemoglobin preferentially over haptoglobin-haemoglobin and is, at the mRNA level, preferentially expressed in epimastigotes (*Jackson et al., 2015*). This, together with the evolutionary history of the trypanosomes, suggests that the receptors from *T. vivax* and *T. congolense* represent the ancestral form, while the *T. brucei* receptor has adopted a modified function and cellular distribution.

The location of *T. congolense* epimastigotes in the tsetse fly mouthparts provides the parasite with the opportunity to acquire nutrients from bloodmeals obtained by the fly. The presence of TcHpHbR will allow epimastigotes to scavenge haem that is present in haemoglobin molecules released from lysed erythrocytes, or to bind more weakly to haptoglobin-haemoglobin. It is noteworthy that the predominant form of bovine haptoglobin adopts higher order multimers more complex than the dimeric HpHb used in this study (*Lai et al., 2007*) and it is possible that this may increase the avidity for TcHpHbR, as observed for multimeric human HpHb binding to scavenger receptor CD163 (*Kristiansen et al., 2001*), making uptake of HpHb more efficient. Endocytosis of TcHpHbR bound to either Hb or HpHb would then provide the parasite with haem.

The expression level of TcHpHbR in the epimastigote form is surprising, with over $5 \times 10^5$ copies per cell. This is approximately 1000-fold greater than the level of TbHpHbR in the bloodstream form (*Vanhollebeke et al., 2008*; *Drain et al., 2001*). One possible explanation for this abundant expression is that the receptor must capture haemoglobin as it periodically and transiently flows over the cell surface when the blood meal passes through the tsetse fly mouthparts. This is in contrast with TbHpHbR expressed in the *T. brucei* bloodstream stage, which will be constantly exposed to its ligand. It remains unknown how *T. congolense* bloodstream forms acquire haem. It may be that sufficient haptoglobin-haemoglobin enters the cell through fluid phase endocytosis or perhaps through an alternative receptor, such as the ortholog of the LHR1 haem transpoter utilized by *Leishmania amazonensis* (*Huynh et al., 2012*).

Our knowledge of the structure and function of HpHbR in both *T. congolense* and *T. brucei* allows us to propose an evolutionary history for the changes that took place in the development of the *T. brucei* receptor. First, perhaps to evade toxic components of the blood meal or to avoid niche competition with other trypanosome species in the proboscis, the developmental cycle of *T. brucei* altered, with the epimastigotes adopting a new location in the salivary glands, rather than developing in the mouthparts. As no haemoglobin is available from bloodmeals in the salivary glands, the

receptor became redundant. A new pattern of expression then evolved in which the receptor was expressed in bloodstream forms instead of in epimastigotes. This switch conferred the ability to more efficiently acquire haem more efficiently in the bloodstream form. That this provides an advantage is evidenced by the attenuation of growth of a TbHpHbR null mutant in a mouse model (*Vanhollebeke et al., 2008*). However, free haemoglobin is not normally present in blood, where it rapidly assembles into HpHb complexes (*Wada et al., 1970*; *Deiss and Lee, 1999*). Evolutionary changes therefore took place in TbHpHbR that resulted in an increase in its affinity for HpHb, from 8 µM in *T. congolense* to 1 µM in *T. brucei* (*Higgins et al., 2013*). A decrease in expression levels and a change in primary location from the cell surface to being concentrated in the flagellar pocket (*Vanhollebeke et al., 2008*) was most likely a final adaptation, perhaps driven by the need to avoid detection by the mammalian acquired immune system.

The change in the developmental stage of expression also had significant effects on the structure of the receptor. On the epimastigote surface, TcHpHbR is free to interact with its ligand relatively unimpeded by other surface proteins. In addition, with a 3 nM affinity for Hb, monovalent binding will allow efficient uptake, and simple tilting of the receptor around its GPI-anchor will position the binding site to allow simultaneous binding of two receptors to one Hb if required (*Figure 7*). A switch to expression in the bloodstream form forced the receptor to function within the densely packed VSG layer. In addition, switching to a ligand with approximately 1000-fold weaker binding made bivalent binding important for efficient uptake. To operate in this new context, the *T. brucei* receptor evolved by gaining both a novel C-terminal domain that probably increases the distance of the ligand binding site from the plasma membrane and by evolving a significant kink between the ligand-binding site and the membrane surface (*Lane-Serff et al., 2014*; *Stødkilde et al., 2014*). This kink pushes the VSG molecules apart and presents the ligand-binding site at the surface. It also allows two receptors, both coupled to the membrane surface, to simultaneously bind to a single dimer of haptoglobin-haemoglobin, increasing avidity and uptake efficiency. A consequence of these changes was the loss of haemoglobin binding, which was no longer under positive selection.

The evolution of the receptor has continued as some primates have acquired innate immune factors that kill trypanosomes. These trypanolytic factors, TLF1 and TLF2, exploit TbHpHbR to increase the efficiency of TLF uptake into *T. brucei* (*Bullard et al., 2012*). For the majority of African trypanosomes this has had a minor effect on parasite survival as non-primates, which lack TLFs, are their predominant hosts (*Hoare, 1972*). However, in *T. brucei gambiense*, the one subspecies that has evolved to infect humans as the main host, the receptor has responded to this new selection pressure through a point mutation that reduces affinity for TLF1 and HpHb (*Higgins et al., 2013*; *DeJesus et al., 2013*; *Symula et al., 2012*).

Therefore, the haptoglobin-haemoglobin receptor of African trypanosomes has undergone a remarkable set of adaptations in its co-evolution with its hosts. It has changed from an epimastigote-expressed haemoglobin receptor into a haptoglobin-haemoglobin receptor, expressed in the blood-stage of *T. brucei* and has adapted to function efficiently in its new surface environment. With an important role at the host-parasite interface, and as a target of innate immunity, it continues to evolve and adapt, allowing it to provide the parasite with a source of haem, while evading destruction by innate immunity factors.

## Materials and methods

### Cloning of *T. vivax* HpHbR

The gene TvY486 0024220 (tritrypdb.org) was identified as the closest homologue to *T. congolense* HpHbR using Blastp. The putative N-terminal signal sequence cleavage site was identified using SignalP (*Petersen et al., 2011*) and the putative C-terminal GPI-anchor addition site was identified by comparison with other trypanosome GPI-anchored proteins. A coding sequence for the mature polypeptide open reading frame plus a Tobacco Etch Virus (TEV) protease site at the N-terminus was synthesised with codons optimised for expression in *E. coli* using the manufacturer's software (Geneart, Thermo Fisher Scientific, Waltham MA) and was cloned into the NdeI and BamHI sites of pET15b.

## HpHbR expression and purification

The *T. b. brucei* HpHbR N-terminal domain and *T. congolense* HpHbR had been previously cloned for expression into a modified pET-15b to include N-terminal hexahistidine tags and cleavage sites for TEV protease (*Lane-Serff et al., 2014*; *Higgins et al., 2013*). To prepare receptors suitable for coupling to a surface plasmon resonance chip, sequences encoding biotin acceptor peptides (BAP) were cloned onto the C-termini of TbHpHbR, TcHpHbR and TvHpHbR.

All three receptors were expressed in *E. coli* Origami B cells. These were induced with 1 mM IPTG (Melford, UK) and incubated for 3 hr at 30°C for TcHpHbR and TvHpHbR, and overnight at 18°C for TbbHpHbR. The protein was purified by $Ni^{2+}$-NTA affinity chromatography, followed by gel filtration using a Superdex 75 16/60 column (GE Healthcare, UK) in 20 mM HEPES pH 7.5, 150 mM NaCl. Protein used in crystallography experiments was cleaved overnight with His-tagged TEV protease at 4°C in 20 mM sodium phosphate pH 7.4, 150 mM NaCl, 3 mM oxidised glutathione, 0.3 mM reduced glutathione to remove the N-terminal His-tag. This was followed by reverse $Ni^{2+}$-NTA affinity chromatography prior to gel filtration.

## HpSP expression, and HpHb complex purification

The SP domain of human haptoglobin had been previously cloned into a modified pAcGP67A vector to generate a polypeptide with an N-terminal hexahistidine tag and a cleavage site for TEV protease. This was expressed in Sf9 insect cells and purified by $Ni^{2+}$-NTA affinity chromatography and gel filtration as described previously (*Lane-Serff et al., 2014*). Full length, dimeric haptoglobin 1–1 was purchased (Sigma Aldrich, St Louis, MO). To purify haemoglobin, human blood was sonicated, followed by anion exchange chromatography using a Mono Q column (GE Healthcare). HpHb and HpSPHb were assembled and purified as described previously (*Lane-Serff et al., 2014*).

## Analytical gel filtration

The assembly of complexes containing TcHpHbR and HpSPHb was assessed using analytical gel filtration chromatography. 0.2 mg of TcHpHbR and 0.3 mg of HpSPHb were mixed (~4:3 molar ratio) before loading onto a Superdex 200 10/300 GL column (GE Healthcare). This was run using an ÄKTApurifier (GE Healthcare) in 20 mM HEPES pH 7.5, 150 mM NaCl.

## Trypanosome cell culture

*T. congolense* IL3000 bloodstream form cells were grown in TcBSF-1 media at 37°C with 5% $CO_2$. *T. congolense* IL3000 procyclic form cells were grown in TcPCF-3 media at 27°C with 5% $CO_2$ (*Coustou et al., 2010*). *T. congolense* procyclic and epimastigote form cells derived from isolates Gam 2 and WG81 were grown in Cunningham's medium (CM) at 27°C. In order to generate epimastigotes, procyclic form cultures were maintained at stationary phase by replacing half of the culture medium every three to four days (*Coustou et al., 2010*). Differentiation to epimastigotes occurred in these cultures after 1–3 months. Epimastigotes were identified by adherence to the culture flask and repositioning of the kinetoplast from posterior and distal to the nucleus, to a position either proximal or anterior to the nucleus. Attempts to harvest the adherent epimastigotes using a cell scraper resulted in damaged/destroyed cells. Some epimastigotes could be dislodged into the supernatant by washing the flask several times with the culture supernatant. Western blots were therefore carried out on the supernatant of these cultures containing both trypomastigote and epimastigote forms.

*T. brucei* TbHpHbR KO bloodstream form cells (*Lane-Serff et al., 2014*) were grown in HMI-9 with 10% FCS at 37°C with 5% $CO_2$ (*Hirumi and Hirumi, 1989*). The TcHpHbR was inducibly overexpressed in *T. brucei* TbHpHbR KO BSFs transfected with pSMOX (*Poon et al., 2012*) and a modified version of pDEX777 (also *Poon et al., 2012*) where the GFP ORF was replaced with the TcHpHbR ORF. Cells were induced with 10 µg/ml doxycycline for 24 hr before protein was harvested and analysed by western blot.

## Tsetse fly infection and dissection

Experimental tsetse flies (*Glossina pallidipes)* were caged in groups of 15, kept at 25°C and 70% relative humidity, and fed on sterile defibrinated horse blood supplemented with 1 mM dATP (*Galun and Margalit, 1969*) via a silicone membrane. Male and female flies were used for

experiments, being given the infective bloodmeal for their first feed 24–48 hr post-eclosion. The infective feed contained approximately $1 \times 10^6$ *T. congolense* Gam 2 trypanosomes $ml^{-1}$ from the supernatant of epimastigote cultures in washed red blood cells supplemented with 10 mM L-gluta-thione (*MacLeod et al., 2007*) to increase infection rates.

Flies were dissected 40–42 days post infection. Heads were removed and proboscides dissected directly into a drop of vPBS on assay slides, carefully separating apart the labrum, hypopharynx and labium. Whole tsetse alimentary tracts were dissected and the proventriculus and midgut placed into separate drops of vPBS.

## Analysis of TcHpHbR protein expression in *T. congolense* epimastigotes

Western blot analysis was carried out on cell lysates using standard methods. Bloodstream and pro-cyclic form cell lysates were harvested from *T. congolense* Il3000 cells from log-phase cultures. Epi-mastigote cell lysates were collected from three independently generated epimastigote-containing cultures.

Antibodies were raised by injecting recombinant TcHpHbR into rabbits (Covalab, France) and purified using affinity chromatographywith TcHpHbR agarose.

Quantification of the copy number of TcHpHbR was carried out by western blot and comparison between cell lysates and recombinant protein. To determine the number of cells expressing TcHpHbR in the samples, six IFAs were carried out (as described below) and 500 cells per IFA were scored as positive or negative for TcHpHbR expression. A total of 50/3000 cells, or 1.67% of the population, were positive for TcHpHbR expression. By comparison with known quantities of recom-binant TcHpHbR protein on two independent western blots it was observed that $8.35 \times 10^4$ TcHpHbR-expressing cells was equivalent to 2.25–4.5 ng protein, or 4.6 and $9.3 \times 10^5$ molecules per cell, using a molecular weight of 35 kDa for calculations.

Immunofluorescent analysis of culture-generated epimastigotes was carried out on culture super-natants as described above or on cells grown on glass coverslips to enrich for the adherent epimasti-gotes. Cells were either fixed with 4% paraformaldehyde at room temperature for 30 min and then blocked with 10 mM methylamine-HCl pH 8.0 for 30 min or fixed by air-drying and then incubating in ice-cold methanol for 30–60 min. Samples were blocked with 5% donkey serum in PBS for 1 hr. Cells were then incubated for 1 hr with rabbit anti-TcHpHbR polyclonal antisera raised against recombinant TcHpHbR protein followed by an Alexa488 donkey anti-rabbit secondary antibody diluted in 5% serum in PBS, also for 1 hr. Cells were stained with 1 µg/ml DAPI for 5 min, washed and mounted with Calbiochem FluorSave Reagent (Merck Millipore, Billerica, MA). For immunofluo-rescent analysis of tsetse-derived *T.congolense*, dissected samples were air-dried and fixed in ice-cold methanol for 30 min, then processed as above. Microscopy was carried out on a Zeiss Imager M1 microscope and analysed with AxioVision Rel 4.8 software.

## Analysis of ligand uptake into *T. congolense* epimastigotes

Hb and BSA were labelled with Alexa Fluor 488 using a protein labelling kit (Thermo Fisher Scien-tific). *T. congolense* WG81 epimastigote-containing cultures (generated as above) were grown on coverslips overnight in serum-free Cunningham's media supplemented with 5 mg/ml BSA, 1 mM hypoxanthine and 0.16 mM thymidine. Incubation in serum-free media was required to remove com-peting Hb ligand from the media. Coverslips were moved onto poly-l-lysine slides and incubated with no ligand, 10 nM Hb-488 or 10 nM-BSA at 27°C for 4 hr. At 2 hr post-addition of ligand, 2 µM protease inhibitor FMK-024 was added. Cells were fixed in 4% paraformaldehyde for 30 min at room temperature, washed 3x in PBS, stained with 1 µg/ml DAPI for 5 min and mounted. Microscopy was carried out as above.

## Crystallisation, data collection and structure determination

HpSPHb and TcHpHbR were mixed in equimolar ratios to a final total concentration of 12.5 mg/ml in 20 mM HEPES pH 7.5, 150 mM NaCl and were subjected to crystallisation trials. Crystals were obtained after 18 hr in a sitting drop format with a well solution containing 0.1 M sodium citrate pH 5, 8% w/v PEG 8000. After cryoprotection by transfer into well solution with the addition of 30% glycerol, the crystals were cryo-cooled. Data were collected on beamline I03 at the Diamond light source and were indexed and scaled using iMosflm (*Battye et al., 2011*) and Scala (*Evans et al.,*

*1993*) respectively. Phaser (*McCoy et al., 2007*) was used to determine a molecular replacement model, using the known structures of TcHpHbR (pdb: 4E40, *Higgins et al., 2013*) and human oxygenated Hb (pdb: 1HHO, *Shaanan, 1983*) as search models. Refinement and rebuilding was completed using Buster (*Bricogne et al., 2011*) and Coot (*Emsley et al., 2010*) respectively.

## Surface plasmon resonance

Receptors were coupled to an SPR chip using a biotin attached to the C-terminal BAP tag. This strategy was designed to allow them to be immobilised with an orientation matching that found on the parasite surface, and to generate a surface that could be readily regenerated. Purified receptors were biotinylated by mixing 1 mg of protein at 30 µM with 20 µg of BirA (Sigma Aldrich), 5 mM ATP (Sigma Aldrich) and 300 µM biotin (Sigma Aldrich). They were incubated at room temperature overnight, before desalting using a PD10 column (GE Healthcare) to remove excess biotin.

SPR experiments were carried out on a Biacore T200 instrument (GE Healthcare). All experiments were performed in 20 mM HEPES pH 7.5, 150 mM NaCl, 0.005% Tween-20 at 25°C. Two-fold dilution series of human Hb, human HpHb or bovine Hb (Sigma) were prepared for injection over a receptor-coated chip with upper concentrations of 1 µM. For each cycle, biotinylated recombinant receptor was immobilised on a CAP chip using the Biotin CAPture Kit (GE Healthcare) to a total loading of ~250 RU. Binding partners were injected for 240 s with a dissociation time of 300 s. The chip was regenerated between cycles using regeneration solution from the Biotin CAPture Kit. The specific binding response of the ligands to receptors was determined by subtracting the response given by Hb or HpHb from a surface to which no receptor had been coupled. As both Hb and HpHb have the capacity to simultaneously interact with two receptors, this data was not fitted to obtain affinity measurements.

## Isothermal titration calorimetry

ITC measurements were carried out on a MicroCal iTC200 System (Malvern, UK). Samples were dialysed for 15 hr into 20 mM HEPES pH 7.5, 150 mM NaCl at 4°C. Experiments were performed at 25°C with 50 µl of *T. congolense* HpHbR at 250 µM titrated into a cell containing 300 µl of Hb at 18 µM. The titrant was injected in 20 injections of 2 µl. Data were integrated and fit by nonlinear least-squares fitting using Origin ITC Software (Malvern).

# Acknowledgements

This work was supported by Medical Research Council Project Grant MR/L008246 to MC and MKH and BBSRC project grant BB/M008924/1 to WG. HL is funded by the Wellcome Trust PhD program in Structural Biology. MKH is a Wellcome Investigator. We thank David Staunton for help with biophysical methods and the beamline scientists at Diamond Light Source for assistance with data collection. We also thank Dr Helen Farr for advice regarding *T. congolense* epimastigote generation in vitro

# Additional information

## Funding

| Funder | Grant reference number | Author |
|---|---|---|
| Wellcome Trust | 101020/Z/13/Z | Harriet Lane-Serff<br>Matthew K Higgins |
| Medical Research Council | MR/L008246/1 | Paula MacGregor<br>Olivia JS Macleod<br>Matthew K Higgins<br>Mark Carrington |
| Biotechnology and Biological Sciences Research Council | BB/M008924/1 | Lori Peacock<br>Christopher Kay<br>Wendy Gibson |

The funders had no role in study design, data collection and interpretation, or the decision to submit the work for publication.

## Author contributions
HL-S, PM, MKH, MC, Conception and design, Acquisition of data, Analysis and interpretation of data, Drafting or revising the article; LP, OJSM, CK, Acquisition of data; WG, Conception and design, Analysis and interpretation of data

## Author ORCIDs
Matthew K Higgins, http://orcid.org/0000-0002-2870-1955

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
