## [Decision Letter]

Thank you for submitting your work entitled "Evolutionary diversification of the trypanosome haptoglobin-haemoglobin receptor" for consideration by *eLife*. Your article has been reviewed by three peer reviewers, and the evaluation has been overseen by Dominique Soldti-Favre as the Reviewing Editor and Richard Losick as the Senior Editor.

The reviewers have discussed the reviews with one another and the Reviewing Editor has drafted this decision to help you prepare a revised submission.

The following individuals involved in the review of your submission have agreed to reveal their identity: Michael Ferguson and Søren Moestrup (peer reviewers).

Summary:

This study convincingly establishes that *Trypanosoma congolense* haptoglobin haemoglobin receptor (TcHpHbR) acts primarily as a high affinity haemoglobin receptor in the epimastigote developmental stage. Surface Plasmon Resonance analysis showed the *T. congolense* and *T. vivax* HpHbR bound both HpHb and Hb but formed a far more stable complex with Hb alone. Based on these observations, the authors suggest that this receptor is tailored for the epimastigote forms to scavenge haemoglobin in the fly. Consistent with a role in the insect vector they found that the HpHbR is not expressed in bloodstream forms but rather in the epimastigote stage found attached to the proboscis. Further, based on IFA and western blot analysis they predict greater than 5 X 105 receptors distributed rather uniformly across the cell surface. Analysis of the structure of the TcHpHbR with bound Hb provided a clear understanding of the features that mediate this high affinity receptor-ligand interaction.

The authors further hypothesize that this ancestor receptor diverged during *T. brucei* evolution giving rise to the HpHb binding properties of the *T. brucei* HpHbR. In *T. brucei*, this receptor evolved to gain distance between ligand binding site and the membrane and also a kink to exclude VSG interactions. Importantly the receptor has morphed into a different shape and specificity to preferentially bind HpHb to satisfy the organisms need for iron – with the complication of host-range modulation through *Trypanosoma* lytic factor uptake.

Overall, this is a well-written paper with important findings that add significantly to our understanding of the crucial interplay between trypanosomes and host haemoglobin.

Essential revisions:

1) The title may suggest that this is a phylogenetic study but it is in fact far more than that. The finding that the ancestral receptor binds the Hb tetramer as shown in the beautiful 3D structure is a key observation that should be included. For instance, 'Evolutionary diversification of the trypanosome haemoglobin-haptoglobin receptor from an ancestral haemoglobin receptor'.

2) Abstract, last sentence. This statement concerning the host immune defence against the *T. brucei* has little relevance for the present study and is confusing for the reader not familiar with the complex innate primate immune response. Further, the reason that some species can overcome this immune response and thereby become human infective is due to another line of evolution in the parasites than described here (evolvement of resistance proteins, mutations etc.).

3) Results: subsection “The *T. brucei* haptoglobin-haemoglobin receptor has evolved from a haemoglobin receptor”, first paragraph: The interaction between human Hp and Hb is extremely strong (K_D_ = 10-15). The authors show that TcHpHbR is capable of dissociating the HpSP-Hb complex. However, the HpSP used in these experiments was expressed in insect cells and not processed correctly into α- and β-chains. Consequently, the recombinant HpSP may have a much lower affinity for Hb compared to native Hp. The authors should investigate if TcHpHbR can also dissociate native Hp-Hb. If TcHpHbR is in fact able to dissociate the native Hp-Hb complex, this suggest the TcHpHbR is entirely an Hb receptor and when it encounters an Hp-Hb complex it can wrest Hb from Hp and only take up Hb.

4) Results: subsection "The haptoglobin-haemoglobin receptor of *T. congolense* is expressed in the epimastigote developmental stage”, third paragraph: It would be relevant to show that the in vitro-generated epimastigote forms that express the receptor bind Hb. Further, it would be relevant to investigate whether they take up Hb or use it as a kind of coating as discussed later.

5) Figure 1: Only human Hb is tested to the *T. congolense* receptor. In view of the structural data suggesting that Hb from other species may bind equally well, it would be relevant to include binding data of other species, e.g. rodent and bovine Hb.

6) Cattle and deer have multimeric Hp due to a duplication of the CCP domain. This may allow the Hp-Hb complexes to bind multiple receptors on the surface of the parasites. The authors should comment on how this may affect the claimed preference of TcHpHbR for Hb over Hp-Hb.

7) Results: subsection “The structural basis for haemoglobin binding”, second paragraph: The authors describe that TcHpHbR interacts both the α- and β-subunits of tetrameric Hb. From Figure 4 it is evident that TcHpHbR binds Hb across the dimer-dimer interface and contacts α1/β2 or α2/β1. This is quite interesting because it suggests that TcHpHbR have evolved to recognize only tetrameric Hb or that the Hb tetramer is perhaps formed on the receptor as a result of interaction with separate Hb dimers. The authors should comment on this.

8) Results: subsection “The structural basis for haemoglobin binding”, third paragraph: The propionate side chains of both Hb subunits are recognized by TcHpHbR. A detailed figure showing these interactions should be included in Figure 4.

9) Results: subsection “The structural basis for haemoglobin binding”, fourth paragraph. A model of the Hp-Hb interaction with TcHpHbR is presented in Figure 5. The statement that "Hp is capable of forming a close interaction with TcHpHbR" should be supported by a detailed figure either as a panel in Figure 5 or as supplement. Is this interaction expected to involve similar contacts as those observed in the TbHpHbR-HpHb structures?

10) Western blots showed two anti-TcHpHb reactive bands, neither at the expected size for unmodified TcHpHb. Since the recombinant TcHpHb runs with the lower, less abundant band this would be consistent with glycosylation differences but this is not shown. The reason for concern is that no evidence is presented to show that the upper band is not an unrelated cross-reactive protein.

11) The IFA images clearly show that a sub-population in the cultures react with that anti-TcHpHb. From these images the authors indicate that all cells where kinetoplast repositioning had occurred reacted with the antibody. Unfortunately, the quality of the images and lack of labeling makes it difficult to see the relationship. Perhaps more important is that based on these images alone the authors conclude that TcHpHb is distributed across the cell surface. However, since the cells were alcohol fixed it remains possible that the IFA is also revealing intracellular protein. Experiments to demonstrate cell surface localization should be straightforward.

12) Discussion: The evolution of the trypanosome receptor from a Hb receptor to a Hp-Hb receptor may go in line with the development of an extended role of Hb as a protein protecting against Hb toxicity in mammalians. Furthermore, there is an interesting parallel between the present data and other date indicating the development of the human Hp-Hb receptor CD163 from a phylogenetic older Hb binding CD163 in mammals – and perhaps also from an older similar to the Hb-binding Pit54 protein in birds (last paragraph in discussion, Etzerodt et al. Antiox. Redox Sign. 18: 2254-63 (2013) and Wicher and Fries PNAS 103:4168-73 (2006)).

13) The role of TcHpHb during the life cycle of the parasite should be more extensively discussed.

13.1) In particular, the role of this receptor in haem acquisition by *T. congolense* is proposed but there is no mention of how, in the absence of the TcHpHbR, the bloodstream and procyclic trypanosomes might acquire haem. If the authors porpose that *T. congolense* and *T. vivax* use transferrin receptors for iron, why does *T. brucei* need a transferrin receptor when it should be able to get iron in the form of haem via the HpHb receptor? How do the authors see the Tc and Tv surviving in the fly when they would be dependent on the fly's blood-meal – would that be frequent enough? – some discussion may be warranted.

13.2) The authors discuss the role of the TbHpHbR in susceptibility to trypanosome lytic factors, however, they clearly show that the HpHbR is not expressed in human serum susceptible bloodstream *T. congolense*. This might be elaborated on.

---

## [Author Response]

Essential revisions:

*1) The title may suggest that this is a phylogenetic study but it is in fact far more than that. The finding that the ancestral receptor binds the Hb tetramer as shown in the beautiful 3D structure is a key observation that should be included. For instance, 'Evolutionary diversification of the trypanosome haemoglobin-haptoglobin receptor from an ancestral haemoglobin receptor'.*

We have changed the title as suggested to ‘Evolutionary diversification of the trypanosome haptoglobin-haemoglobin receptor from an ancestral haemoglobin receptor’.

2) Abstract, last sentence. This statement concerning the host immune defence against the T. brucei has little relevance for the present study and is confusing for the reader not familiar with the complex innate primate immune response. Further, the reason that some species can overcome this immune response and thereby become human infective is due to another line of evolution in the parasites than described here (evolvement of resistance proteins, mutations etc.).

As suggested, we have removed the final sentence of the Abstract.

3) Results: subsection “The T. brucei haptoglobin-haemoglobin receptor has evolved from a haemoglobin receptor”, first paragraph: The interaction between human Hp and Hb is extremely strong (K_D_ = 10-15). The authors show that TcHpHbR is capable of dissociating the HpSP-Hb complex. However, the HpSP used in these experiments was expressed in insect cells and not processed correctly into α- and β-chains. Consequently, the recombinant HpSP may have a much lower affinity for Hb compared to native Hp. The authors should investigate if TcHpHbR can also dissociate native Hp-Hb. If TcHpHbR is in fact able to dissociate the native Hp-Hb complex, this suggest the TcHpHbR is entirely an Hb receptor and when it encounters an Hp-Hb complex it can wrest Hb from Hp and only take up Hb.

At the start of the Results section of our manuscript we describe an experiment in which we mixed *T. congolense* HpHbR with human HpSPHb complex and observed this to result in the formation of a complex of TcHpHbR with Hb and free HpSP. We describe this experiment as it was the observation that led to our discovery that TcHpHbR is an Hb receptor, rather than to imply that TcHpHR will disassemble HpHb as part of its physiological function. As the reviewers rightly point out, the HpSPHb used in this experiment is not identical to native HpHb complex, as native Hp is cleaved into α and β chains. While structural alignments show that this does not affect the position of residues involved in the interaction with the haptoglobin-haemoglobin receptors, it does alter the interface between Hp and Hb and could change the stability of the HpHb complex. We therefore performed the experiment suggested by the reviewers in which we incubated HpHb with TcHpHbR, and in this case we do not see disassembly of the HpHb. However, neither do we see the formation of a TcHpHbR:HpHb complex with sufficient stability to survive gel filtration chromatography. This is in contrast to what occurs when we mix TcHpHbR with Hb, when a stable complex is formed. This data is now presented in Figure 1—figure supplement 1. This supports our conclusion that TcHpHbR is primarily an Hb receptor and also shows that it is unlikely that TcHpHbR can disassemble HpHb during the acquisition of haem.

*4) Results: subsection "The haptoglobin-haemoglobin receptor of T. congolense is expressed in the epimastigote developmental stage”, third paragraph: It would be relevant to show that the in vitro-generated epimastigote forms that express the receptor bind Hb. Further, it would be relevant to investigate whether they take up Hb or use it as a kind of coating as discussed later.*

The reviewers asked for further experiments in which we test the functional consequences of the expression of the receptor to see whether *T. congolense* epimastigotes can internalise haemoglobin or whether a coat of haemoglobin assembles on the parasite surface. Culture-derived epimastigotes were therefore incubated with fluorescently-labelled human haemoglobin. These data (now presented in Figure 4) show clear internalisation of the labelled haemoglobin into epimastigotes, while there is no internalisation into trypomastigotes. In contrast, we did not observe any accumulation of labelled haemoglobin on the cell surface in these experiments. We have therefore confirmed that the *T. congolense* epimastigotes can use TcHpHbR to allow them to internalise haemoglobin.

5) Figure 1: Only human Hb is tested to the T. congolense receptor. In view of the structural data suggesting that Hb from other species may bind equally well, it would be relevant to include binding data of other species, e.g. rodent and bovine Hb.

The reviewers point out that the original manuscript showed only the binding of TcHpHbR to human haemoglobin. As *T. congolense* is not human infective, they ask for confirmation that it also binds to haemoglobins from other species. In the original manuscript we presented a figure (Figure 4—figure supplement 1) showing that the haemoglobin residues that contact TcHpHbR are absolutely conserved between human and bovine haemoglobin. In this revised version we make two additions. We show experimentally that TcHpHbR does indeed interact with bovine Hb, with an affinity similar to that of human Hb, and we describe this data in the fourth paragraph of the subsection “The *T. brucei* haptoglobin-haemoglobin receptor has evolved from a haemoglobin receptor”. This is presented in Figure 1—figure supplement 2). We also increase the range of species (all known to be infected by *T. congolense*) included in Figure 4—figure supplement 1, showing that the residues contacted by *T. congolense* are conserved in cows, sheep, horses and camels.

6) Cattle and deer have multimeric Hp due to a duplication of the CCP domain. This may allow the Hp-Hb complexes to bind multiple receptors on the surface of the parasites. The authors should comment on how this may affect the claimed preference of TcHpHbR for Hb over Hp-Hb.

The reviewers mention that Hp from bovine and deer adopt higher order multimers than those present in the dimeric HpHb used in this study, and that this might increase the efficiency of uptake of HpHb. This is possible, although it is also possible that the arrangement of the binding sites in the larger HpHb complexes might not allow increased binding to receptors associated with the parasite surface for steric reasons. We have added a sentence in the second paragraph of the Discussion to raise this issue for the reader.

*7) Results: subsection “The structural basis for haemoglobin binding”, second paragraph: The authors describe that TcHpHbR interacts both the α- and β-subunits of tetrameric Hb. From Figure 4 it is evident that TcHpHbR binds Hb across the dimer-dimer interface and contacts α1/*β*2 or α2/*β*1. This is quite interesting because it suggests that TcHpHbR have evolved to recognize only tetrameric Hb or that the Hb tetramer is perhaps formed on the receptor as a result of interaction with separate Hb dimers. The authors should comment on this.*

The reviewers ask whether the structure suggests that the receptor only interacts with a haemoglobin tetramer or whether two haemoglobin dimers can assemble together on the receptor. In the absence of affinity measurements for the haemoglobin dimer (which would require a stabilised haemoglobin dimer) we cannot present the quantitative analysis that we would have liked to include. However, we speculate that either is possible in the second paragraph of the subsection “The structural basis for haemoglobin binding“.

8) Results: subsection “The structural basis for haemoglobin binding”, third paragraph: The propionate side chains of both Hb subunits are recognized by TcHpHbR. A detailed figure showing these interactions should be included in Figure 4.

As suggested, we have remodelled the original Figure 4, now Figure 5, removing panels B and E and replacing them with close up views of the interactions that TcHpHbR makes with the haem groups of the haemoglobin α- and β-chains.

9) Results: subsection “The structural basis for haemoglobin binding”, fourth paragraph. A model of the Hp-Hb interaction with TcHpHbR is presented in Figure 5. The statement that "Hp is capable of forming a close interaction with TcHpHbR" should be supported by a detailed figure either as a panel in Figure 5 or as supplement. Is this interaction expected to involve similar contacts as those observed in the TbHpHbR-HpHb structures?

To illustrate our model of the TcHpHbR:HpHb complex we have added a second view, taken from the bottom of the receptor, to Figure 6 (originally Figure 5). This will show, in more detail, how close the loops of Hp come to the receptor in this configuration. In addition, the interaction between TbHpHbR and Hp in the TbHpHbR:HpHb complex is mostly hydrophobic in nature and we now point out that a similar, but smaller, hydrophobic patch is present in TcHpHbR, centred around Phe48, which can mediate an equivalent hydrophobic interaction with Hp. We prefer not to show the model in further detail (for example showing the side chains) as this would be an over-interpretation of the model. Instead it is presented to support our hypothesis that the β-subunit of Hb will interact with the upper binding site on TcHpHbR when it binds to the HpHb complex.

10) Western blots showed two anti-TcHpHb reactive bands, neither at the expected size for unmodified TcHpHb. Since the recombinant TcHpHb runs with the lower, less abundant band this would be consistent with glycosylation differences but this is not shown. The reason for concern is that no evidence is presented to show that the upper band is not an unrelated cross-reactive protein.

The reviewers also asked for some further experiments to confirm that the bands recognised by anti-TcHpHbR are not reacting with an unrelated protein. This comment was stimulated by the higher molecular weight of the antibody reactive band in the epimastigotes when compared with recombinant protein. In the absence of an epimastigote form of a *T. congolense* HpHbR^-/-^ cell line, we tested this using *T. brucei* TbHpHbR^-/-^ bloodstream forms carrying an inducible copy of a TcHpHbR transgene. Upon induction with doxycycline, the antisera detected two bands that again ran at a higher molecular weight than the recombinant TcHpHbR. This data is presented in Figure 2—figure supplement 1 and shows that a fraction of *T. brucei* expressed TcHpHbR runs in the same position as the antibody-reactive band in the epimastigotes. This difference in mobility is probably due to differential modification of GPI-anchors in the bloodstream and insect cell stages of *T. brucei*. The insect stages are modified with sialic acid residues whereas the bloodstream forms, used here for the transgene expression, are not. In conjunction with our new data showing haemoglobin uptake into *T. congolense* epimastigotes, this gives strong support to our conclusion that TcHpHbR is expressed in epimastigotes.

*11) The IFA images clearly show that a sub-population in the cultures react with that anti-TcHpHb. From these images the authors indicate that all cells where kinetoplast repositioning had occurred reacted with the antibody. Unfortunately, the quality of the images and lack of labeling makes it difficult to see the relationship. Perhaps more important is that based on these images alone the authors conclude that TcHpHb is distributed across the cell surface. However, since the cells were alcohol fixed it remains possible that the IFA is also revealing intracellular protein. Experiments to demonstrate cell surface localization should be straightforward.*

To confirm that the TcHpHbR is surface localised, we have collected further IFA images of epimastigotes, both culture-derived and dissected from tsetse flies. We have also conducted imaging experiments using paraformaldehyde-fixed, non-permeabilised cells. We see clear surface staining in both the permeabilise and non-permeabilised cells, clearly indicating surface localisation. We also confirm this in tsetse-derived epimastigotes, showing that receptor expression in eptimastigotes is not a consequence of in vitro differentiation. This data is now presented in a revised version of Figure 3.

12) Discussion: The evolution of the trypanosome receptor from a Hb receptor to a Hp-Hb receptor may go in line with the development of an extended role of Hb as a protein protecting against Hb toxicity in mammalians. Furthermore, there is an interesting parallel between the present data and other date indicating the development of the human Hp-Hb receptor CD163 from a phylogenetic older Hb binding CD163 in mammals – and perhaps also from an older similar to the Hb-binding Pit54 protein in birds (last paragraph in discussion, Etzerodt et al. Antiox. Redox Sign. 18: 2254-63 (2013) and Wicher and Fries PNAS 103:4168-73 (2006)

We thank the referees for pointing out the interesting parallels in changes in ligand specificity between published findings on CD163 and our findings. We have mentioned these in the Results section (subsection “The *T. brucei* haptoglobin-haemoglobin receptor has evolved from a 149 haemoglobin receptor”, fourth paragraph) and by the addition of Etzerodt et al. in the references section. While the changes in the specificity of CD163 are interesting, we do not believe that there is evidence to suggest that these changes are related to the changes in specificity of HpHbR in their functional consequence, as alterations in HpHbR specificity correlated with a change in life cycle stage usage.

13) The role of TcHpHb during the life cycle of the parasite should be more extensively discussed.

13.1) In particular, the role of this receptor in haem acquisition by T. congolense is proposed but there is no mention of how, in the absence of the TcHpHbR, the bloodstream and procyclic trypanosomes might acquire haem. If the authors porpose that T. congolense and T. vivax use transferrin receptors for iron, why does T. brucei need a transferrin receptor when it should be able to get iron in the form of haem via the HpHb receptor? How do the authors see the Tc and Tv surviving in the fly when they would be dependent on the fly's blood-meal – would that be frequent enough? – some discussion may be warranted.

We agree with the reviewers that the switch in expression stage of HpHbR leads to a number of questions. We have speculated in the manuscript (Discussion, third paragraph) that the high expression level of TcHpHbR is present to allow capture of Hb from the blood meal as it flows through the mouthparts of the tsetse fly and that lower expression levels are acceptable in the mammalian bloodstream where the parasite is more continuously exposed to HpHb and so does not need a transient capture mechanism. This adaptation might allow the parasite to acquire sufficient haem during its time in the tsetse fly, but an experimental demonstration of this will be challenging. It is also possibility that there are other mechanisms by which *T. congolense* acquires haem in the mammalian bloodstream, as not discussed in the aforementioned paragraph.

13.2) The authors discuss the role of the TbHpHbR in susceptibility to trypanosome lytic factors, however, they clearly show that the HpHbR is not expressed in human serum susceptible bloodstream T. congolense. This might be elaborated on.

We have also implied that the susceptibility to trypanolytic factors will have been increased as a result of the switch in expression of HpHbR to the blood stream in our Discussion (for example in the sixth paragraph). However, the other methods for HpHbR-independent uptake of trypanolytic factors into trypanosomes mean that we prefer not to discuss this point at length.